Identifying DUSP-1 and FOSB as hub genes in immunoglobulin A nephropathy by WGCNA and DEG screening and validation

Jianping Wu
Wei Xiaona
Li Jiajia
Zhang Rui
Han Qianqian
Yang Qiongqiong yangqq@mail.sysu.edu.cn
Department of Nephrology, Sun Yat Sen Memorial Hospital , Sun Yat-sen University, Guangzhou , China
Liu Jinhui
Electronic publication date: 2022 Jul 25
Publication date: 2022
Volume: 10
Electronic Location ID: e13725
Received 2022 Feb 14; Accepted 2022 Jun 22
Copyright: © 2022 Jianping et al.
Copyright year: 2022
Copyright holder: Jianping et al.
License: This is an open access article distributed under the terms of the Creative Commons Attribution License, which permits unrestricted use, distribution, reproduction and adaptation in any medium and for any purpose provided that it is properly attributed. For attribution, the original author(s), title, publication source (PeerJ) and either DOI or URL of the article must be cited.
License URL: https://creativecommons.org/licenses/by/4.0/

Keywords: Immunoglobulin A nephropathy, Weighted gene co-expression network, Hub genes, Differentially expressed genes

Funding: National Key R&D Programmes of China 2021YFC2009400 Science and Technology Projects in Guangzhou 201904010142 Natural Science Foundation of Guangdong Province 2021A1515010801 This work was supported by the National Key R&D Programmes of China (2021YFC2009400); the Science and Technology Projects in Guangzhou (201904010142); and the Natural Science Foundation of Guangdong Province (2021A1515010801). The funders had no role in study design, data collection and analysis, decision to publish, or preparation of the manuscript.

==============================
Background

The mechanism of immunoglobulin A nephropathy (IgAN) is still unknown. A bioinformatics analysis is a powerful method to identify the biomarkers and possible therapeutic targets of a certain disease from related datasets.

Methods

The GSE93973 dataset, obtained from the Gene Expression Omnibus (GEO) database, was used to construct a weighted gene co-expression network (WGCNA) and filter differentially expressed genes (DEGs). The biological process (BP) enrichment among all the genes in the key modules was analyzed through a Gene Ontology (GO) enrichment analysis. We selected the overlap of hub genes in the WGCNA and Protein-Protein Interaction (PPI) network as the final hub genes in IgAN. We verified the final hub genes in two other datasets and in clinical kidney tissue specimens. A receiver operating characteristic (ROC) curve was used to evaluate the diagnostic efficacy of hub genes for IgAN.

Results

The turquoise module, which contained 1,806 genes, was the module with the highest correlation coefficient with IgAN in the GSE93973 dataset. The GO enrichment analysis showed that these 1,806 genes were mainly enriched in inflammation and immune responses. There were five hub genes identified by WGCNA and 34 hub genes identified in a DEG analysis in the GSE93973 dataset. DUSP1 and FOSB were identified as the final hub genes in IgAN. The validation results of the final hub genes in two other databases and clinical kidney tissue specimens validated the result that, compared to the control group, FOSB and DUSP1 were expressed at lower levels in the glomerulus of IgAN patients. The ROC curve indicated that DUSP1 and FOSB were good diagnostic indicators for IgAN.

Conclusions

Our analysis identified two hub genes that might be potential targets for the intervention and treatment of IgAN.

Introduction

Immunoglobulin A nephropathy (IgAN), the most common primary glomerulonephritis, is characterized by the presence of IgA-containing immune complexes in the mesangial area of the glomerulus. Most IgAN patients will progress to end-stage kidney disease (Lai et al., 2016; Woo et al., 2010). The pathogenesis of IgAN is not very clear and there are not currently any effective pharmacological treatments. There is, therefore, an urgent need to identify key molecular targets involved in the intervention and treatment of IgAN.

A weighted gene co-expression network analysis (WGCNA) is a useful bioinformatics method of identifying and clustering genes related to diseases or clinical manifestations and forming different co-expression networks or co-expression modules (Langfelder & Horvath, 2008). WGCNA is also a powerful data analysis tool that can identify potential biomarkers for the intervention and treatment of different diseases, such as advanced kidney disease (Xin et al., 2020), Sjögren’s syndrome (Yao et al., 2019), acute myocardial infarction (Wu et al., 2021), and pancreatic cancer (Zhou et al., 2019).

In this study, we identified two IgAN hub genes in the GSE93798 dataset using WGCNA and DEG screening. The flowchart of the study is shown in Fig. 1. The final hub genes were validated in the GSE26316 and GSE104948 datasets and in kidney tissue from IgAN patients and kidney cancer adjacent sample tissues. This study identified two hub genes, Dual Specificity Phosphatase 1 (DUSP1) and FosB Proto-Oncogene (FOSB), that might be promising targets for the intervention and treatment of IgAN.

Figure 1 Flowchart of this research.

Materials and Methods

Data extraction

We performed our analysis on August 1, 2021. The GSE93798 dataset (https://www.ncbi.nlm.nih.gov/geo/query/acc.cgi?acc=GSE93798), which includes the gene matrix of the glomerular compartment of renal biopsies from 20 healthy people and 22 IgAN patients, was acquired from the Gene Expression Omnibus (GEO) database. The two groups of people in dataset GSE93798 were with gender and age matched. IgAN (age 43 ± 15 years old, women-to-men ratio of 1:2) and control samples (age 48 ± 13 years old, women-to-men ratio of 1:1.27). This dataset was then analyzed using WGCNA and DEG screening to identify key modules and hub genes related to IgAN. The datasets were based on the platform of GPL22945 (Affymetrix Human Genome U133 Plus 2.0).

Construction of co-expression network

To select the top 5,000 median absolute deviation (MAD) genes for WGCNA, we sorted all gene expression matrices of the GSE93798 dataset according to median deviation size using the R package “WGCNA” (https://cran.r-project.org/web/packages/WGCNA/index.html) (Langfelder & Horvath, 2008). A cluster analysis was performed on the samples before constructing the WGCNA network to test the heterogeneity between the samples included in the GSE93798 dataset. The “PickSoftThreshold” package (https://www.rdocumentation.org/packages/WGCNA/versions/1.71/topics/pickSoftThreshold) was used to determine the best soft threshold and to identify the co-expression module based on the dynamic cut method. The module and sample feature correlation heat map and module and sample feature hierarchical clustering map were constructed to determine which key module had the highest correlation coefficient with IgAN. Genes with a gene significance (GS) value and module membership (MM) that were both greater than 0.9 were determined to be hub genes.

Functional enrichment analysis

Visualization and Integrated Discovery (DAVID) (https://david.ncifcrf.gov/home.jsp/) was used to perform an online Gene Ontology (GO) analysis within the genes in the key module. The “GOplot” R package (https://www.rdocumentation.org/packages/GOplot/versions/1.0.2) was then used to plot and display the results.

DEG screening

Genes with an adjusted P-value <0.05 and |log2(fold change)| > 1 were defined as DEGs between the control group and IgAN patients. The GSE93798 dataset was screened using the “limma” package (https://www.rdocumentation.org/packages/limma/versions/3.28.14) from R (Ritchie et al., 2015). All DEGs were drawn as heat maps with R package “pheatmap.” (https://www.rdocumentation.org/packages/pheatmap/versions/1.0.12) The Cytoscape software was used to construct a Protein-Protein Interaction (PPI) network based on the protein interaction information obtained from the STRING database (https://string-db.org/) after screening the DEGs (Li et al., 2015). The Molecular Complex Detection (MOCDE) plugin was then applied to find hub genes in the PPI network.

Validation of final hub genes in other datasets and clinical specimens

The hub genes that existed in both the PPI network and WGCNA were determined to be the final hub genes. In order to find appropriate datasets, based on the GEO database, we only included datasets that included: (1) Sequencing data from glomerus of IgAN patients; (2) gene matrix information; (3) and 3 or more samples. Two datasets met the criteria and were included, namely GSE37460 (https://www.ncbi.nlm.nih.gov/geo/query/acc.cgi?acc=GSE37460), which contains the renal biopsies of 27 healthy people and 27 IgAN patients, and GSE104948 (https://www.ncbi.nlm.nih.gov/geo/query/acc.cgi?acc=GSE104948), which contains the renal biopsies of three healthy people and 27 IgAN patients. To validate these hub genes, we verified their expressions in three cases of adjacent renal cell carcinoma tissues and three cases of IgAN renal biopsy tissues with immunohistochemistry. The verified specimens came from Sun Yat-sen Memorial Hospital of Sun Yat-sen University. This study was approved by the Medical Ethics Committee of Sun Yat-sen Memorial Hospital (SYSEC-KY-KS-2018-080). All patients included in the study signed an informed consent form.

Receiver operating characteristic (ROC) curve analysis

The ROC curve of the final hub genes was drawn using the sample information in the GSE93798, GSE37460 and GSE104948 datasets with the pROC package (https://www.rdocumentation.org/packages/pROC/versions/1.18.0/topics/pROC-package) of R (Langfelder & Horvath, 2008). We evaluated the diagnostic efficacy of the final hub genes with IgAN by calculating the area under the curve (AUC).

Immunohistochemical staining

After dewaxing, the paraffin sections with a thickness of about 4 μm were placed in a sodium citrate solution with pH = 7.0 heated in a microwaves on high heat for antigen retrieval. A 3% hydrogen peroxide solution was then dropped on the sliced tissue to block endogenous peroxidase. We then performed primary antibody incubation (anti-DUSP1, Abcam, ab61201, 1:100 dilution; anti-FOSB, Abcam, ab11959, 1:100 dilution), HRP goat anti-rabbit secondary antibody (1:200 dilution) incubation, and nucleus staining incubation. These were followed by DAB coloration, hematoxylin staining, and hydration, at which point we covered the slip with neutral resin and observed the final product under a microscope. We then used the Image J software to perform a semi-quantitative analysis.

Statistics

A statistical analysis was performed using R 4.0.3, SPSS 20.0 and GraphPad Prism 6.0. A Student’s t-test was then performed to examine the statistical difference between the group means. All data were presented as Mean ± SEM. P < 0.05 was defined as statistically significant.

Results

Construction of the WGCNA network

All gene expression matrices of the GSE93798 dataset were sorted according to median deviation size and the top 5,000 genes were used for subsequent analyses. A cluster analysis was performed on the samples before we started constructing the WGCNA network and the results showed that there was no obvious heterogeneity between these samples (Fig. 2A). The power value (β) was set to 8 and the scale independence value was set to 0.85 (Figs. 2B and 2C). There was a negative correlation between the logarithm log(k) and the logarithm log(P(k)), and the correlation coefficient was greater than 0.8. These results proved that the constructed co-expression network met the requirements of scale-free topology (Figs. 2D and 2E). Therefore, β = 8 was chosen to build a hierarchical clustering tree.

Figure 2 Data verification and filter the best soft threshold.

(A) Sample clustering to view heterogeneity. (B) Soft threshold non-scale fitting index analysis. (C) Soft threshold average connectivity analysis. (D) Histogram of the connectivity distribution when the soft threshold value was 8. (E) Check scale-free topology scale when the soft threshold value was 8.

Identifying the key module and hub genes

There were six gene co-expression modules in the WGCNA (Fig. 3A). The turquoise module, which contained 1,806 genes, was the one with the highest correlation coefficient (CC) with IgAN (Figs. 3B and 3C). A total of 400 genes were randomly selected for heat map display (Fig. 3D). The module and disease characteristic heat map showed that the turquoise module had the most significant correlation with IgAN (Fig. 3E). Genes that met the criteria that the value of GS and MM were both greater than 0.9 were determined to be hub genes (Fig. 3F).

Figure 3 The co-expression network constructed based on the GSE93798 dataset.

(A) Gene dendrogram. (B) Heat map of module–trait relationships. (C) Distribution of average gene significance in the six modules. (D) Heat maps visualizing 400 randomly selected genes in the network to depict the TOM. (E) The combination of eigengene dendrogram and heatmap. (F) The distribution of GS and MM was presented as a scatter diagram in the turquoise module.

Functional enrichment analysis of turquoise module genes

A GO enrichment analysis showed that these 1,806 genes in the turquoise module were mainly enriched in the BP of cell adhesion, leukocyte migration, inflammatory response, extracellular matrix organization, immune response, and adaptive immune response (Fig. 4A). A cross-check GO analysis showed that a large number of genes related to white blood cell migration, inflammation and inflammatory response were also enriched in extracellular matrix tissue, immune response, adaptive immune response and other BPs. This indicated that these cross-mapped genes were involved in the pathogenesis of IgAN in multiple BPs (Fig. 4B).

Figure 4 GO analysis of the key module genes.

(A) The top six components of the GO analysis for biological processes. (B) Phylogenic tree of the results of the GO analysis for biological processes.

DEGs screening

There were 248 DEGs identified, including 107 up-regulated genes and 141 down-regulated genes, after screening the DEGs based on the GSE93798 dataset (Figs. 5A and 5B). A total of 34 hub genes in three clusters were screened using the MCODE method when the threshold of k-score > 2. The three clusters included 15 nodes (Fig. 5C), 11 nodes (Fig. 5D) and 18 nodes (Fig. 5E), respectively.

Figure 5 DEG screening.

(A, B) Volcano and heatmap of DEGs by screening the GSE93798 dataset. (C, D, E) Gene cluster filtered using the MCODE plugin. (F) The final hub genes determined by Venn diagram.

Identification and verification of final hub genes

Hub genes found in both the WGCNA and PPI network, namely FOSB and DUSP1, were identified as the final hub genes (Fig. 5F). Compared with the control group, FOSB and DUSP1 had low expression levels in the glomerulus of IgAN patients in the GSE93798 dataset and the two validation datasets, GSE104948 (Fig. 6A) and GSE37460 (Fig. 6B). To validate our results in clinical specimens, we used immunohistochemistry and semi-quantitative analyses, which showed that compared with adjacent tissues of renal cancer, FOSB and DUSP1 were down-regulated in the glomerulus of IgAN patients. This result was consistent with the expressions of these two genes seen in the transcriptome matrix data (Figs. 6C–6E).

Figure 6 Validation of key genes.

(A, B) Expression of FOSB and DUSP1 in the validation datasets of GSE37460 (A) and GSE104948 (B). (C) Representative image of FOSB and DUSP1 immunohistochemistry in renal biopsy tissues of the control group (n = 3) and the IgAN group (n = 3). (D, E) The semi-quantitative analyses of C. An asterisk (*) indicates P < 0.01.

ROC curve analysis of final hub genes

The ROC curve was drawn to evaluate the diagnostic value of FOSB and DUSP1 using the sample information in the GSE93798, GSE37460 and GSE104948 datasets (Fig. 7). All the AUC values were greater than 0.75, indicating their potential value as diagnostic markers for IgAN.

Figure 7 The ROC curve of FOSB and DUSP1 in datasets.

The ROC curve of FOSB and DUSP1 in the GSE37460 (A), GSE93798 (B) and GSE104948 (C) datasets; ROC, receiver operating characteristic; AUC, area under the ROC curve.

Discussion

IgAN is one of the main causes of end-stage renal disease worldwide. The pathogenesis of IgAN is unclear, but the current hypothesis is the multi-strike theory that various pathogenic factors lead to the increase of galactose-deficient O-glycan (Gd-IgA1) in the glomerulus, activating the mesangial cells, and ultimately promoting the occurrence and development of IgAN (Lai et al., 2016). According to this hypothesis, specific glomerular biological information could better identify the hub genes of IgAN, which supported our decision to use datasets that included glomerulus samples from IgAN patients in this study rather than those that only included peripheral blood samples (Liu et al., 2020; Jiang et al., 2020).

The co-expression network of diseases and modules constructed by WGCNA can identify biomarkers and possible therapeutic targets for a certain disease. In this study, we used WGCNA to identify the key module with the highest correlation coefficient with IgAN in the GSE93798 dataset. A GO enrichment analysis showed that the genes in the key module were mainly enriched in the BP of cell adhesion, leukocyte migration, inflammatory response, extracellular matrix organization, immune response, and adaptive immune response. This result indicated that the pathogenesis of IgAN is related to the activation of immune and inflammation-related pathways, but the inflammatory response plays a more important role in the development of IgAN than the immune response. IgAN is currently considered an autoimmune disease (Suzuki et al., 2011), but Sheng et al. (2018) found that Notch Receptor 1 (NOTCH 1) and NF-κB and other inflammatory pathway-related molecules were increased in the renal tissue of IgAN patients. Some animal experiments have also shown that certain drugs can affect the progression of IgAN by inhibiting NF-κB signaling and NLRP3 inflammasome activation (Bai et al., 2019). In clinical studies, some scholars have found that the neutrophil to lymphocyte ratio is an independent risk factor for poor prognosis in IgAN patients (Li et al., 2020). Inflammation thus plays an important role in the occurrence and development of IgAN. This may partly explain why immunosuppressive agents are ineffective in treating IgAN (Rauen et al., 2020).

To find the most significant hub gene, we defined GS > 0.9 and ME > 0.9 as target values in the key module of WGCNA and used MCODE to identify the most important gene clusters in the PPI network. There were five hub genes in the WGCNA analysis and 34 hub genes in the MCODE analysis. The final hub genes found in both the WGCNA and MCODE analysis were defined as the key hub genes; FOSB and DUSP-1 were identified as the final hub genes of IgAN. An ROC analysis showed that FOSB and DUSP-1 had good diagnostic performance. The validation results of these key genes in two other datasets, as well as in clinical specimens, were consistent with the results that the expression of FOSB and DUSP1 were down-regulated in the renal glomerulus of IgAN patients compared with people that did not have IgAN. Previous studies have shown that FOSB is related to human stress and is a potential therapeutic target for depression. FOS family gene polymorphisms, including FOSB, are related to the clinical phenotypes of IgAN patients (Park et al., 2014). In an animal model of renal ischemia/reperfusion, the expression of FOSB was up-regulated in interthalamic core cells (Mutoh, Ohsawa & Hisa, 2013). Liao et al. (2020) found that MicroRNA-27A-3P targeted FOSB to regulate cell function involved in the occurrence and development of IgAN. These studies show that FOSB is involved in the progression of IgAN, but the specific mechanism of FOSB in the pathogenesis of IgAN unclear. DUSP1 can regulate cell death, tumor growth and plays a role in both inflammation and chronic metabolic diseases. DUSP1 is highly expressed in tissues such as the heart and liver that could activate mitogen-activated protein kinase (MAPKs), extracellular signal-regulated kinase (ERK), c-Jun N-terminal kinase (JNK), and p38 to promote cell apoptosis (Johnson & Lapadat, 2002). Jin et al. (2018) showed that DUSP1 inhibited cardiac ischemia-reperfusion (IR) injury by regulating downstream JNK to inhibit mitochondrial fission. Low DUSP1 expression is related to glucose metabolism disorder, renal dysfunction and glomerular apoptosis; by affecting mitochondrial homeostasis, DUSP1 is able to protect against diabetic nephropathy progression (Sheng et al., 2019; Ge et al., 2019). There is currently no research related to DUSP1 in IgAN and its potential role in the pathogenesis of IgAN needs further study. In this study, we found two genes closely related to IgAN using bioinformatics and in vitro verification. These genes can be used as potential markers for the diagnosis of IgAN, and perhaps also as potential therapeutic targets.

This research has several highlights. First, we used datasets that include glomeruli from the biopsies of IgAN patients for our analysis in order to better identify candidate biomarkers and therapeutic targets, contributing to a more comprehensive understanding of IgAN. Second, WGCNA is a powerful tool for bioinformatics analysis. It can associate gene modules with disease traits. WGCNA plus a DEG analysis can find the most relevant value target genes. However, due to limited sample availability we were unable to verify the mRNA expression levels of FOSB and DUSP1. Instead, we analyzed kidney tissue slices using immunohistochemistry and achieved consistent results with the bioinformatics analysis.

Conclusions

In summary, our study identified FOSB and DUSP1 as final hub genes in IgAN through WGCNA and DEG analysis. These genes were verified to be significantly down-regulated in the renal glomerulus of IgAN patient kidney tissues and other IgAN datasets, compared with non-IgAN subjects. The specific mechanisms of these hub genes involved in IgAN pathogenesis also need to be further studied. Our results provided novel molecular targets as biomarkers and therapeutic targets for future research.

Supplemental Information

Supplemental Information 1 Raw data for expression of DUSP1 clinical tissue samples (Figure 6D).

Click here for additional data file.

Supplemental Information 2 Raw data for expression of FOSB a clinical tissue samples (Figure 6D).

Click here for additional data file.

Thanks to Dr. Anyi Wang for polishing the manuscript.

Additional Information and Declarations

Competing Interests

Author Contributions

Human Ethics

Data Availability

The authors declare that they have no competing interests.

Wu Jianping performed the experiments, prepared figures and/or tables, authored or reviewed drafts of the article, and approved the final draft.

Xiaona Wei performed the experiments, prepared figures and/or tables, authored or reviewed drafts of the article, and approved the final draft.

Jiajia Li analyzed the data, prepared figures and/or tables, authored or reviewed drafts of the article, and approved the final draft.

Rui Zhang analyzed the data, prepared figures and/or tables, authored or reviewed drafts of the article, and approved the final draft.

Qianqian Han analyzed the data, prepared figures and/or tables, authored or reviewed drafts of the article, and approved the final draft.

Qiongqiong Yang conceived and designed the experiments, prepared figures and/or tables, authored or reviewed drafts of the article, and approved the final draft.

The following information was supplied relating to ethical approvals (i.e., approving body and any reference numbers):

This study was approved by the Medical Ethics Committee of Sun Yat-sen Memorial Hospital (SYSEC-KY-KS-2018-080).

The following information was supplied regarding data availability:

The data is available at NCBI GEO: GSE123568, GSE26316, GSE104948.

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
