# Peer review of "Identifying DUSP-1 and FOSB as hub genes in immunoglobulin A nephropathy by WGCNA and DEG screening and validation"

_PeerJ, doi:10.7717/peerj.13725_

## Round 0.1 · original submission · Major Revisions

It is my opinion as the Academic Editor for your article - Identifying DUSP-1 and FOSB as hub genes in immunoglobulin a nephropathy by WGCNA and DEGs screening and validation - that it requires a number of Major Revisions.

Reviewer 1 ·

Basic reporting

no comments

Experimental design

no comments

Validity of the findings

no comment

Additional comments

WU et al. found key 2 key genes related to IgA nephropathy through WGCNA analysis and differential analysis using GEO database, and verified their expression through other data sets and clinical samples. However, this paper still has the following deficiencies:
1. The author used a total of three data sets for analysis. Why does the WGCNA analysis and difference analysis only focus on data set GSE93798? The author claimed to have found two key genes related to IgA nephropathy, but the analysis of these two genes was based on data set GSE93798. Then, are these two genes also key genes related to IgA nephropathy in the other two data sets?
2. The author's description of the immunohistochemical process is too simple. It only introduces the source of primary antibody, but does not introduce the thickness of wax block section, what tissue repair solution is used, the source and concentration of secondary antibody, and the number of immunohistochemical repetitions.
3. In fact, the author only analyzed functional enrichment without pathway analysis, which is inconsistent with the description in the paper. Why did the author only enrich the genes obtained by WGCNA analysis while not enrich the genes obtained by difference analysis when conducting functional enrichment? At the same time, Figure 4A and Figure 4B are the enrichment results, one is the enrichment results of the first 6, the other is the enrichment results of the first 9, whether there is duplication?
4. The numbering of pictures in Fig. 5 should be carried out in sequence. Why there is only IgAN in Fig. 5B, and what blue represents needs to be explained. Fig. 6 About the Immunohistochemical results, there were 2 images of each gene in the control group and IgAN group respectively, while the author used 3 pairs of samples, which need further explanation. At the same time, the control group of patients was normal during data set analysis, but the control group was para-cancer samples of patients with renal cancer when clinical samples were used for verification. Please explain. In Fig. 6, the E is shown in the annotation, but not found in the picture. The AUC of FOSB gene of GSE93798 in Fig. 7 is significantly lower than 1, however in this figure, the AUC is marked as 1. Please explain.
5. There are many punctuation and formatting errors in the article, and grammar polishing is needed.

·

Basic reporting

IgAN is a common health issue that needs to be understood. The authors used bioinformatics approaches to identify key hub genes for IgAN and validated their findings in clinical samples. This will provide critical insight for other researchers to further study this disease. The results in the manuscript are well-related to the hypothesis.

However, this manuscript is poorly written in unprofessional English with numerous grammar problems and missing parts. This needs to be corrected before the manuscript can be accepted.

For example,
in line 27: specimens. used receiver operating characteristic (ROC) curve to evaluate the diagnostic efficacy. (missing the subject)

In line 123: Set the power value (β) to 8 and the scale independent value was 0.85.(Figure.2B, C),There was negative correlation between the logarithm log(k) and the logarithm log(P(k) and the correlation coefficient is greater than 0.8, c. (wrong uppercase, a space is needed after a period, the last "c" is meaningless.)

Other small errors also need to be corrected, such as using "et al." instead of "et al".

Experimental design

The experiment in the manuscript is well-designed. The authors used one dataset to draw the conclusion and validated it in another two datasets and clinical samples.

Additional information about the methods needs to be provided: Since the analysis is using publicly available tools, the authors need to provide the date when they performed their analysis, as well as the version of code/package/software they used for their analysis. This is essential for other researchers to replicate the study in the manuscript.

Validity of the findings

The conclusion is well stated in the manuscript. The authors provide sufficient samples to validate their findings.

A minor point: in line 222 all the databases that include glomerulus tissue of IgAN to make bioinformatics analysis as glomerular biological information analysis could better identify candidate biomarkers and therapeutic targets contributing to a more comprehensive understanding of IgAN. (This is an overstatement since the author needs to justify that ALL the databases including glomerulus have been used in this study. This is almost impossible)

---

## Round 0.2 · Major Revisions

The authors do not solve reviewer 1's problem.

Reviewer 1 ·

Basic reporting

no comments

Experimental design

no comments

Validity of the findings

no comments

Additional comments

This article provides a good idea, and the author has a strong ability of bioanalysis and basic experiments, but the author does not seem to be very serious. Such as, after revision, in Figure 7A, it is clear that THE AUC of DUSP1 is higher than that of FOSB, but the notation shows that DUSP1 is lower than FOSB.

·

Basic reporting

IgAN is a common health issue that needs to be understood. The authors used bioinformatics approaches to identify key hub genes for IgAN and validated their findings in clinical samples. This will provide critical insight for other researchers to further study this disease. The results in the manuscript are well-related to the hypothesis.

The authors have modified the manuscript with improved written English. The manuscript is clear and easy to understand.

There are still some minor errors that need to be corrected:
line 172: and the two validation datasets, GSE104948 (Figure 6A) and GSE37460 (Figure B). The second figure label should be (Figure 6B)

Experimental design

The experiment in the manuscript is well-designed. The authors used one dataset to draw the conclusion and validated it in another two datasets and clinical samples.

Following the suggestion from the reviewer, the author provided the date and versions for all datasets and methods that are used for this manuscript.

Validity of the findings

The conclusion is well stated in the manuscript. The authors provide sufficient samples to validate their findings.

The author corrected the statement regarding glomerulus databases.

---

## Round 0.3 · accepted · Accept

I am writing to inform you that your manuscript - Identifying DUSP-1 and FOSB as hub genes in immunoglobulin A nephropathy by WGCNA and DEG screening and validation - has been Accepted

Reviewer 1 ·

Basic reporting

no comment

Experimental design

no comment

Validity of the findings

no comment

Additional comments

The author has carefully revised it.